# Printed Directional Bending Sensor with High Sensitivity and Low Hysteresis for Human Motion Detection and Soft Robotic Perception

**DOI:** 10.3390/s23115041

**Published:** 2023-05-24

**Authors:** Yi-Fei Wang, Ayako Yoshida, Yasunori Takeda, Tomohito Sekine, Daisuke Kumaki, Shizuo Tokito

**Affiliations:** Research Center for Organic Electronics (ROEL), Yamagata University, 4-3-16, Jonan, Yonezawa 992-8510, Yamagata, Japan

**Keywords:** flexible bending sensor, printed electronics, conductive composite, wearable sensor, soft robotics

## Abstract

We present a high-performance flexible bending strain sensor for directional motion detection of human hands and soft robotic grippers. The sensor was fabricated using a printable porous conductive composite composed of polydimethylsiloxane (PDMS) and carbon black (CB). The utilization of a deep eutectic solvent (DES) in the ink formulation induced a phase segregation between the CB and PDMS and led to a porous structure inside the printed films after being vapored. This simple and spontaneously formed conductive architecture provided superior directional bend-sensing characteristics compared to conventional random composites. The resulting flexible bending sensors displayed high bidirectional sensitivity (gauge factor of 45.6 under compressive bending and 35.2 under tensile bending), negligible hysteresis, good linearity (>0.99), and excellent bending durability (over 10,000 cycles). The multifunctional applications of these sensors, including human motion detection, object-shape monitoring, and robotic perceptions, are demonstrated as a proof-of-concept.

## 1. Introduction

Flexible strain sensors or bending sensors have garnered attention for their use in motion capture and gesture recognition of both humans and robots and have proven to be a valuable tool in human behavior study and robot control [1,2,3,4]. Compared to conventional optical camera systems, these flexible sensors have advantages in simple structure, compact size, and less susceptibility to environmental interference. Furthermore, flexible sensors are often more cost-effective and flexible than the sensors based on microelectromechanical systems (MEMS), making them a preferred choice for integration into robots and wearable devices [5,6]. Flexible bending sensors have been developed using various sensing principles, such as piezoelectric, capacitive, and piezoresistive [7,8,9]. Piezoelectric sensors exhibit excellent sensitivity and low power consumption, but they are limited by their material systems and are only capable of dynamic detection [7,10]. Capacitive and piezoresistive sensors, on the other hand, can detect both dynamic and static stimuli. Of these two, piezoresistive sensors are superior in sensitivity, have a simple structure, and are easy to integrate into circuits [11,12,13,14]. Therefore, piezoresistive-type flexible sensors are ideal for practical applications.

Conductive composites have received extensive research attention as piezoresistive-type flexible strain sensors due to their tunable mechanical and electrical properties [15,16]. In these composites, conductive fillers such as silver nanoparticles (AgNW), carbon nanotubes (CNT), carbon black (CB), and graphite are dispersed within a polymer matrix, creating a random conductive pathway that is flexible and stretchable [17,18,19,20,21]. However, conventional conductive composites have significant limitations, including high hysteresis and low linearity, which negatively impact their accuracy and reliability in practical applications [21,22,23,24]. Additionally, few of them could provide directional information on motion detection, requiring the use of multiple sensors in a sensing network. To overcome these challenges, researchers have proposed constructing a porous conductive network by decorating the porous wall with conductive fillers, leading to higher sensitivity and reduced hysteresis [25,26,27,28,29]. However, the fabrication of this porous conductive structure is a complex process, often requiring the use of sacrificial templates or conductive solution dips, making it difficult to pattern and scale. Hence, a flexible strain sensor with the capability of directional bending sensing, high sensitivity, low hysteresis, and scalable fabrication is highly desired.

Recently, we developed a self-segregated porous PDMS/CB conductive composite using a deep eutectic solvent (DES) [30,31]. This composite was produced through a simple mixing and one-step printing process, eliminating the need for complex materials synthesis and post-treatment. The phase segregation between CB and PDMS results in high sensitivity to pressure and stretch with minimal hysteresis, making it a highly desirable material for flexible bending sensors. This study presents a fully stencil-printed flexible directional bending sensor that employs the self-segregated porous conductive composite and thoroughly investigates its electromechanical response to small strains induced by bending. In contrast to our previous stretchable strain sensor, which was developed to detect large deformation strains (~100%) through stretching [30], the present sensor is designed for flexible interface applications to detect tiny bending strains (−0.6% ~ +0.6%). Importantly, this sensor can detect both the magnitude and direction of strain, surpassing the capabilities of our previous stretchable strain sensor and conventional flexible bending sensors, which are limited to detecting strain in only one direction (typically tensile strain). This advantage of distinguishing bending direction will greatly reduce the number of sensors required within the sensor network, which will largely reduce the complexity of their large-scale applications, such as soft robotic skins. Our results indicate that this porous composite-based sensor outperforms the sensor made from a random composite in terms of directional bending sensing performance, with high sensitivity (gauge factor of 45.6 under compressive bending and 35.2 under tensile bending), low hysteresis, good linearity (R^2^~0.99), and excellent stability (over 10,000 cycles). Demonstrations of human motion detection, shape monitoring of a flexible substrate, and gesture monitoring of a soft robot gripper reveal the sensor’s promising potential in multifunctional mechanosensory applications.

## 2. Materials and Methods

### 2.1. Materials

From Teijin DuPont Films (Tokyo, Japan), 100-μm thick poly(ethylene-phthalate) (PEN) film (Q65HA) was purchased. PDMS (Sylgard 184) was purchased from Dow Corning (Midland, MI, USA). Benzophenone (BP) and Diphenylamine (DP) were purchased from TCI Chemicals (Tokyo, Japan). Carbon Black (CB) (Ketjenblack EC600JD) was purchased from Lion Specialty Chemicals (Tokyo, Japan). Screen-printable Ag paste (XA-3609) was purchased from Fujikura Kasei (Tokyo, Japan). All materials were used as received.

### 2.2. Ink Preparation

Composite inks were prepared according to our previous reports [30,31], with slight modifications. As shown in Appendix A, BP and DP were mixed with a molar ratio of 1:1 to achieve a deep eutectic liquid. The mixed solid was stored at room temperature for 2 h or annealed at 90 °C for 30 min to obtain a clear yellowish liquid. A proper amount of carbon black was added to the DES and mixed by a planetary centrifugal mixer (THINKY MIXER AR-100) for 7 min (5 min mixing and 2 min degassing). The achieved DES-CB gel was further mixed with PDMS (base polymer: cure agent 10:1) by the mixer for 15 min to achieve a slurry like ink. In this study, weight ratio of PDMS to DES was fixed as 2:1 in the final ink formulations. The randomly dispersed PDMS/CB composite was also prepared as the reference by mixing the proper amount of carbon black with PDMS (base polymer: cure agent 10:1) in the mixer for 15 min (10 min mixing and 5 min degassing). The detail of ink formulations is provided in Appendix A.

### 2.3. Fabrication of the Printed Sensors

Sensors were fabricated using a simple stencil printing method [30,31]. A polyimide film tape (OKAMOTO 1030WD) was patterned with a plotter cutter (Silhouette Cameo 4) with a digital pattern designed by AutoCAD. Then the film was attached to the PEN substrate as a mask for stencil printing. The ink was added to the masking film’s head and the blade for a squeegee made of glass. Electrodes were printed by using the silver paste and sintered at a temperature of 150 °C for 30 min. We then printed the composite ink and performed an annealing process of 75 °C for 1 h and 140 °C for 30 min. The reference devices of randomly dispersed PDMS/CB were printed and annealed under the same conditions. The geometry of the printed sensor is shown in Appendix A. The fabricated sensor was cut to the required size and attached with a copper wire (50 μm) as a connection for subsequent characterization and measurements.

### 2.4. Characterization 

An optical image of the printed trace and substrate was observed with an optical microscope (Keyence VHX-7000), and the thickness of different printed layers was measured using an Olympus LEXT OLS4100 3D laser scanning microscope. SEM images were obtained using a HITACHI tabletop microscope (TM4000Plus). Photos of the sensors demonstrated on a human-hand flexible film and a soft robotic gripper were taken using a smartphone camera. The bending sensing tests were conducted by applying a high-precision electric slider (EZS, oriental motor) with software (MEXE02). A digital multimeter recorded the resistance (KEITHLEY DMM6500) with 2 wire measurements. For the demonstration of human motion monitoring, the developed sensors were attached to the hand of a volunteer (Male, 33 years old). The experiments for monitoring the hand gesture of a volunteer using the developed sensing device were approved by Yamagata University’s institutional review board (R3-38).

## 3. Results and Discussion

### 3.1. Design and Fabrication of Flexible Bending Sensor

There are three key considerations to designing a high-performance flexible directional bending sensor using a conductive composite. Firstly, the composite must possess high anisotropic sensitivity to both tensile and compressive strain. This means that the composite must exhibit large resistance increases under tension and decreases under compression, which allows the sensor to easily distinguish the bending direction (Figure 1a). Secondly, low mechanical-electric hysteresis is essential to achieve high accuracy and reliability in detecting the tiny strain induced by bending. Finally, easy and scalable fabrication is crucial for transitioning the sensor from laboratory to real-world applications. Unfortunately, many efforts to develop flexible sensors tend to prioritize high parameter numbers or new materials systems while disregarding practical considerations. This has made the large-scale application of flexible sensors a challenge. Our recent studies have revealed that a porous conductive architecture can be easily achieved with just a simple mixture of deep eutectic solvent (DES), PDMS, and CB (Appendix A). The composite film can be fabricated using a one-step stencil printing and annealing process, which is both simple and scalable (see Section 2, Appendix A). The porous structure and segregation between the PDMS and CB induced by the DES result in a composite that exhibits both high sensitivities to stretch and pressure and excellent low hysteresis. Therefore, this porous PDMS/CB conductive composite is an ideal match for the requirements of a high-performance flexible bending sensor.

The device structure of our sensor is depicted in Figure 1b. It is constructed on a PEN substrate, with Ag serving as the electrode and a porous conductive composite as the sensing layer. The fabrication process is detailed in Figure 1c, wherein stencil printing is employed due to its low cost, scalability, high speed, and compatibility with high-viscosity composite inks. A polyimide film mask is digitally designed via plot cutting, and the silver paste electrode is printed first on the PEN substrate, followed by the sensing layer using our composite ink. The geometry of the printed sensor is demonstrated in Appendix A. The DES and annealing process is crucial for the formation of the self-segregated porous conductive structure for the sensing layer, as illustrated in Appendix A. The van der Waals and π–π interactions between DES and carbon materials lead to the CB having a stronger affinity for dispersing in the DES than in PDMS, causing spontaneous phase separation [30,31,32,33]. During pre-annealing (75 °C for 1 h), the DES/CB domains come into contact due to sedimentation and establish a conductive pathway. Removal of the DES subsequently (140 °C for 30 min) produces a segregated conductive composite film with a porous structure. We expect that this straightforwardly formed conductive network could offer superior bending sensing performance in printed sensors. An example of the printed sensor is shown in Figure 1d. As a reference, we also fabricated and characterized a sensor using a random PDMS/CB composite.

### 3.2. Characterization of Printed Conductive Composite Films

The conductivity and morphology of the printed composite films were characterized as shown in Appendix A and Figure 2. The porous composites demonstrated relatively high conductivity and low percolation threshold (CB loading of 1.92 wt%) when compared with the random composite (CB loading of 3.94 wt%). Specifically, at a CB loading of 4 wt%, the porous composite exhibited a conductivity of 3.7 × 10^−2^ S/cm, while the random composite showed a low conductivity of 4.0 × 10^−6^ S/cm at the same CB loading. This enhanced conductivity in the porous composite is attributed to the phase segregation of PDMS and CB, which creates sufficient conductive pathways compared to the randomly dispersed composite. In conventional conductive composites, there exists a trade-off between conductivity and flexibility, as high conductivity typically requires a high loading of conductive fillers, leading to a reduction in the mechanical softness of the composite. Our porous conductive composite overcomes this challenge by achieving higher conductivity at lower CB loadings. This provides distinct advantages, such as the ease of adjusting sensor resistance and circuit design, along with a softer material when compared to the random composite. The cross-sectional SEM images of the printed conductive composite films are presented in Figure 2. At a CB loading of 2 wt%, the porous composite displayed a clear 3D elastic frame of PDMS with CB aggregates inside the pores, forming conductive pathways (Figure 2a,b). Increasing the CB loading to 4 wt% did not affect the phase segregation between the CB and PDMS, but it reduced the size of the aggregated CB conductive domain and void space in the pores, as shown in Figure 2c,d. In contrast, the random composite did not show any phase segregation or presence of pores, which may limit its sensitivity to both strain and compression, as shown in Figure 2e,f. Considering the film structure of the printed conductive composites, a high bending sensitivity can be expected in porous composites due to the aggregation of the CB conductive network and the void space in pores. Under tensile strain, the aggregated CB conductive pathway resulted in a significant increase in resistance, while under compressive strain, the void space allowed the conductive network to be easily compressed, leading to a significant decrease in resistance. Therefore, it is expected that the porous composite with a CB loading of 2 wt% will show the highest bending sensitivity compared to that of 4 wt% and the random composite. Additionally, it is worth noting that increasing the carbon loading in the random composite resulted in higher ink viscosity, which made it difficult to process and produced a relatively rough surface morphology. According to a previous study, phase segregation between the conductive filler and polymer matrix results in low mechanical and electrical hysteresis in composites. This is because it reduces the interaction and friction between the nano/micro-sized filler particles and polymer chains [30]. Therefore, the porous composite demonstrated advantages in terms of conductivity and morphology, which will benefit the design and fabrication of high-performance flexible sensors.

### 3.3. Directional Bending Sensing Performances

The directional bending sensing performance of the flexible sensors based on the printed composites was systematically studied and is shown in Figure 3 and Appendix A. In the experiment setting, the sensing layer (thickness ~110 μm) was printed on a PEN film (thickness ~100 μm), and outward and inward bending was applied to the PEN film. As illustrated in Figure 3a, the displacement of the slide changes the chord length of the sensor, inducing strain in the sensing layer. The strain values can be calculated utilizing the equation *ε = ±h/r*, where *h* denotes the distance of the sensing layer from the neutral axis of the bent substrate, and *r* represents the curvature radius. The relationship between the curvature radius *r* and the chord length *c* is given by *c = 2r sin(l/2r)*, where *l* represents the arc length of the sensor under the bending state [34,35]. In this work, the PEN substrate has an elastic module (~5 Gpa) much higher than that of the printed sensing layer (~0.3 Mpa); we assumed that the bending occurred through the neutral axis of the PEN film. Therefore, the *h* value is determined as 50 μm. The arc length of the sample is 40 mm. During measurement, the chord length of the PEN film was decreased from 40 mm to 10 mm under a speed of 1 mm/s. We first evaluated the resistance changes of printed sensors based on porous conductive composite with different CB loading (Figure 3a and Appendix A). As we expected, the porous PDMS/CB composite with a CB loading of 2wt% showed the best sensing performance in terms of resistance change during the bending state. As shown in Figure 3b, when the displacement increases to 30 mm, the resistance increases by 20% under tensile strain and decreases by 26% under compressive strain, exhibiting adequate sensitivity over the entire measured range. In contrast, the porous conductive composite with a CB loading of 4wt% showed a relatively low sensitivity with a resistance change of only ~5% in the bending measurement (Appendix A). The sensor utilizing the random PDMS/CB composite was evaluated as a reference. However, due to its proximity to the percolation threshold, the random composite sensor with a CB loading of 4 wt% exhibited a large resistance with an unstable bending response, rendering it unsuitable as a reference device. As a result, the random composite with a CB loading of 5 wt% was evaluated, as shown in Figure 3c, and it demonstrated inadequate bending sensitivity.

Electrical hysteresis is another important parameter of sensing performances, which has significant influences on the accuracy and reliability of sensors. However, it is a challenge in the conductive composite-based sensors. To evaluate the hysteresis of our printed devices and conduct a quantitative comparison of the hysteresis magnitude, we defined the degree of hysteresis (*DH*) as Equation (1):*DH* = (*A*_B_ − *A*_R_)/*A*_R_(1)
where *A*_B_ and *A*_R_ are the areas of bending and releasing curves, respectively. As shown in Figure 3b, the sensor based on porous composite (with CB loading of 2 wt%) shows negligible hysteresis of 4.96% and 2.3% for tensile strain and compressive strain, respectively. At the same time, the random PDMS/CB composite (with CB loading of 5 wt%) shows a big hysteresis of 29.4% and 24.8% for tensile strain and compressive strain, respectively. We attribute the reduced hysteresis in porous composite to the phase segregation between PDMS and CB, as we have discussed in the morphology characterization. Moreover, the initial resistance of the porous composite-based sensor (with CB loading of 2 wt%) is around 21 kΩ, while a random PDMS/CB composite-based sensor (with CB loading of 5 wt%) showed a high resistance of 5 MΩ. Based on the above results, the porous composite with CB loading of 2 wt% was selected as the optimized material for the flexible bending sensor and was characterized in detail in Figure 3d–f. To evaluate the bending performance of sensors made from the porous PDMS/CB composite (with CB loading of 2 wt%), the resistance change was plotted as a function of compressive and tensile strain, as shown in Figure 3d. The strain sensitivity, i.e., the gauge factor (*GF*), was calculated using Equation (2):*GF* = Δ*R*/*R*_0_ ε (2)
where Δ*R* is the resistance change of the sensor when bending was applied, R_0_ is the resistance of the sensor without any bending strain induced (0%), and *ε* is the induced strain under bending. The results demonstrate that the sensor exhibits exceptional sensitivity to the bending-induced strain, with a maximum GF of 35.2 to tensile strain and a high *GF* of 45.6 to compressive strain. In addition to the high sensitivity, the relative resistance of the sensor changes linearly according to the applied strain with a remarkable linearity of *R*^2^ > 0.99. For the applications in robotics, a low detection limitation and fast response speed are also desired. As shown in Figure 3e, under a tiny strain of 0.05%, the sensor’s relative resistance changes curve showed a distinct response, indicating notably low detection limits. The response and recovery times of the sensor under tensile strain were evaluated to be 200 ms and 240 ms, respectively. Similarly, under compressive strain, the response and recovery times were determined to be 300 ms and 180 ms, respectively. It is comparable to that of other composite-based sensors, although not superior. Nevertheless, we believe that it is sufficient for monitoring human body motion and soft robotics at the current stage. It is also worth noting that the response time can be further reduced by improving the measurement setup. Our current equipment has limitations in measuring motion speed, which may have impacted the observed response time. Finally, a durability test of the sensor was conducted by subjecting it to 5000 cycles of bending/releasing strain (~0.4%) in each direction. As depicted in Figure 3f, the sensor’s resistance change did not exhibit any notable baseline drift or signal degradation in response, indicating superior mechanical and electrical durability.

Table 1 shows the comparisons of our printed sensors with recent studies [34,36,37,38,39,40]. Compared to previously reported directional bending sensors, which often struggle to achieve high sensitivity in both directions, typically exhibiting high sensitivity for either tensile or compressive strain but low sensitivity in the opposite direction, our study demonstrates the superior performance of our porous composite sensors in terms of sensitivity to both tensile and compressive strains. Moreover, the fabrication method we employed allows for scalability, cost-effectiveness, and reproducibility of the device, making it an attractive option for mass production and practical applications. In contrast to the previous strategies that aim for impressive results but often rely on sophisticated fabrication techniques and expensive materials, our approach prioritizes simplicity and uniformity, essential qualities for integrating the sensor device with supporting circuitry. These findings highlight the importance of considering both performance and fabrication in the development of strain sensors, especially as they continue to be applied in diverse fields such as robotics and wearable healthcare.

### 3.4. Multifunctional Mechanosensory Application

The flexible sensor developed in this study displays exceptional sensitivity, linearity, and low hysteresis, thus presenting a promising solution for multifunctional mechanosensory applications. Figure 4 shows the performance of the sensor in various applications. For instance, when affixed to the second joint of an index finger, the sensor produced distinguishable signals in different finger bending states, which could be utilized for hand gesture recognition (Figure 4a). Similarly, when attached to the wrist, the sensor could detect movement with directional information (Figure 4b). The assembly of multiple sensors on a substrate facilitates the monitoring of changes in the shape of a flexible film, as shown in Figure 4c. Three flexible bending sensors, S1, S2, and S3, were mounted on a flexible PEN film. In the initial state (i), no resistance changes were observed in any of the sensors. When the film was bent to state (ii), S2 showed a significant resistance increase, indicating a large tensile strain applied in the center position. When a large compressive strain was applied in the center (iii), S2 showed a significant resistance decrease. Similarly, in the bent state (iv), the increased resistance of S1 indicated a tensile strain in its position, while the decreased resistance of S3 indicated a compressive strain. By examining the changes in the resistance value of sensors S1, S2, and S3, the bending direction of each part of the flexible film can be detected, enabling the monitoring of the film’s shape. These features prove advantageous for bendable human–machine interfaces, such as flexible displays. With an appropriately designed sensor array pattern, this sensor is also expected to detect the complex motions of flexible interfaces, including twist motions, which will be conducted in our future work. Furthermore, the sensor can be utilized in soft robotic perceptions, as demonstrated in Figure 4d and Appendix A, with the sensor affixed to the strain-limited layer of a soft robot gripper. In this application, the response signal exhibited clear changes as the soft gripper actuated in different bending states. Compared to previous studies that utilized complex and expensive processes, our strain sensor exhibits superior functional properties and can be readily fabricated using printing processes compatible with large-scale patterning and circuit integration, rendering it an attractive option for practical applications.

## 4. Conclusions

We have successfully developed a high-performance flexible bending sensor by utilizing a porous conductive composite. The composite ink was prepared through a simple mixing process of PDMS, CB, and a deep eutectic solvent (DES), and the sensor was fabricated via stencil printing, a simple and scalable technique. Compared to sensors based on conventional random composites, our porous composite sensors possess a unique porous structure and segregation between PDMS and CB, which enables superior mechanical-electric performance. Our developed sensor demonstrates high sensitivity (GF ~ 45.6 for compressive bending, GF ~ 35.2 for tensile bending), negligible hysteresis, good linearity (R^2^ ~ 0.99), and excellent stability (>10,000 cycles). This sensor provides a simple and straightforward approach to realizing a flexible directional bending sensor for multifunctional mechanosensory applications. Its comprehensive performance level makes it suitable for various applications, including human motion monitoring, object shape monitoring, and tactile motion perceptions. Future work is needed to investigate a sensor network with high performance uniformity and integration technologies for sensors on wearable and soft robotic interfaces to realize practical applications. 

## Figures and Tables

**Figure 1 sensors-23-05041-f001:**
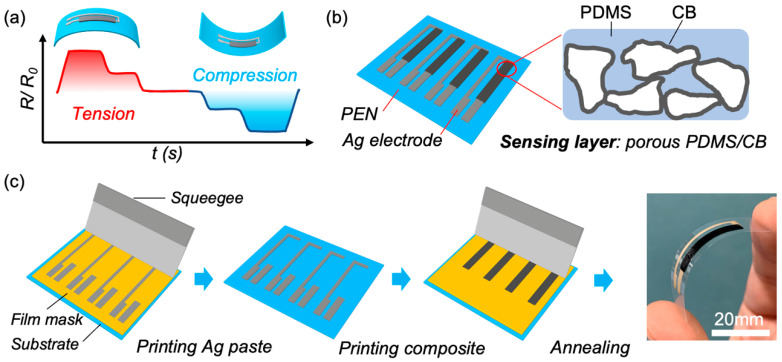
(**a**) Schematic of directional bending sensing through the printed flexible sensor; (**b**) Illustration of device structure and porous PDMS/CB conductive composite; (**c**) Illustration of the fabrication process and photograph of the printed flexible bending sensor.

**Figure 2 sensors-23-05041-f002:**
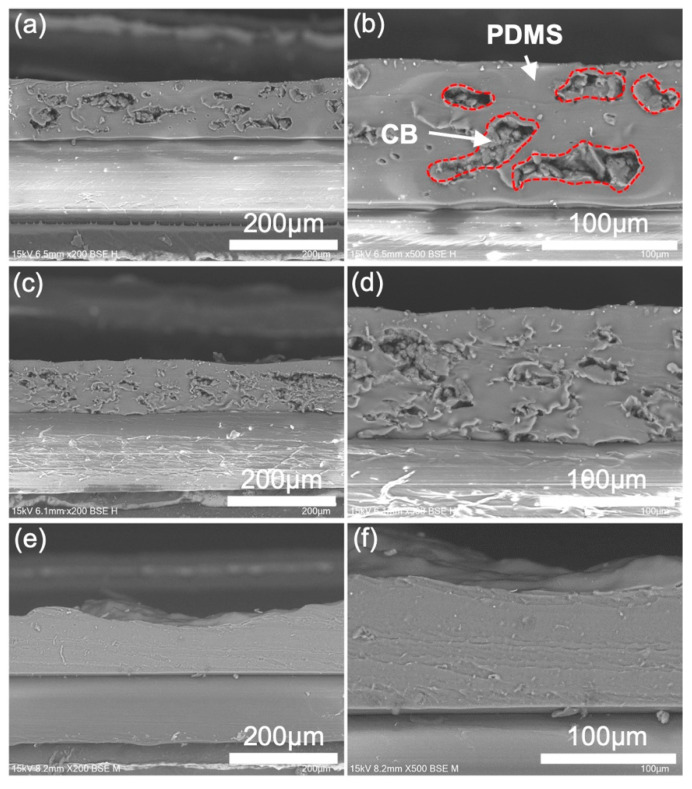
Cross-sectional SEM images of the printed conductive composites. (**a**,**b**) Porous PDMS/CB composite with CB loading of 2 wt%; (**c**,**d**) Porous PDMS/CB composite with CB loading of 4 wt%; (**e**,**f**) Random PDMS/CB composite with CB loading of 5 wt%.

**Figure 3 sensors-23-05041-f003:**
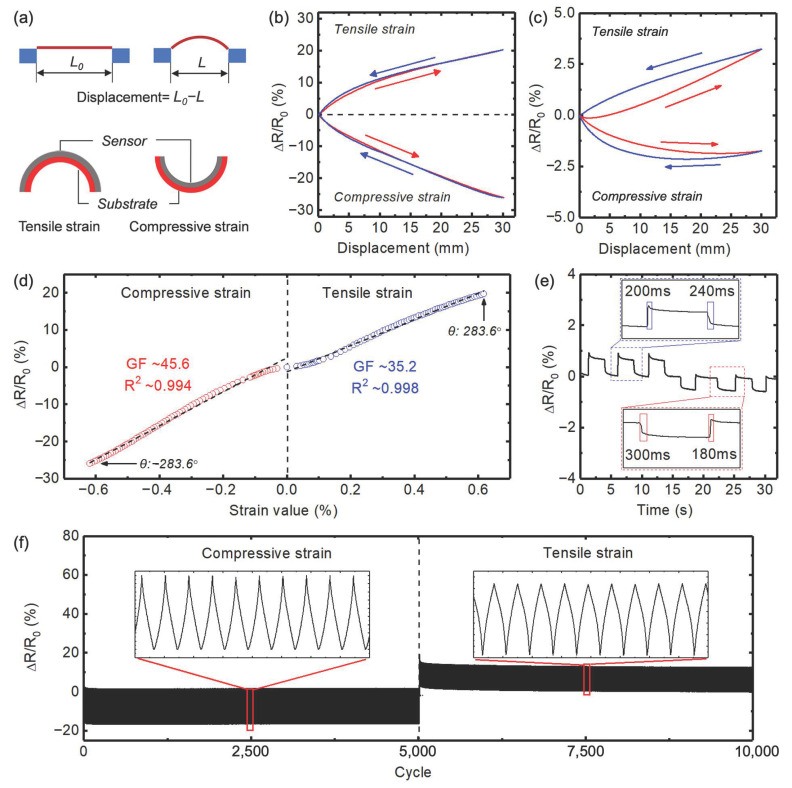
Characterization of directional bending sensing performances. (**a**) Illustration of the characterization setup for flexible bending sensor, with applied compressive and tensile strain; (**b**) Relative resistance changes of porous PDMS/CB composite-based sensor with CB loading of 2 wt% under different bending states; (**c**) Relative resistance changes of random PDMS/CB composite based sensor with CB loading of 5 wt% under different bending states; (**d**) Relative resistance changes to the applied strain of porous PDMS/CB composite based sensor with CB loading of 2 wt%; (**e**) response time of porous PDMS/CB composite based sensor with CB loading of 2 wt%; (**f**) cyclic stability of porous PDMS/CB composite based sensor with CB loading of 2 wt%.

**Figure 4 sensors-23-05041-f004:**
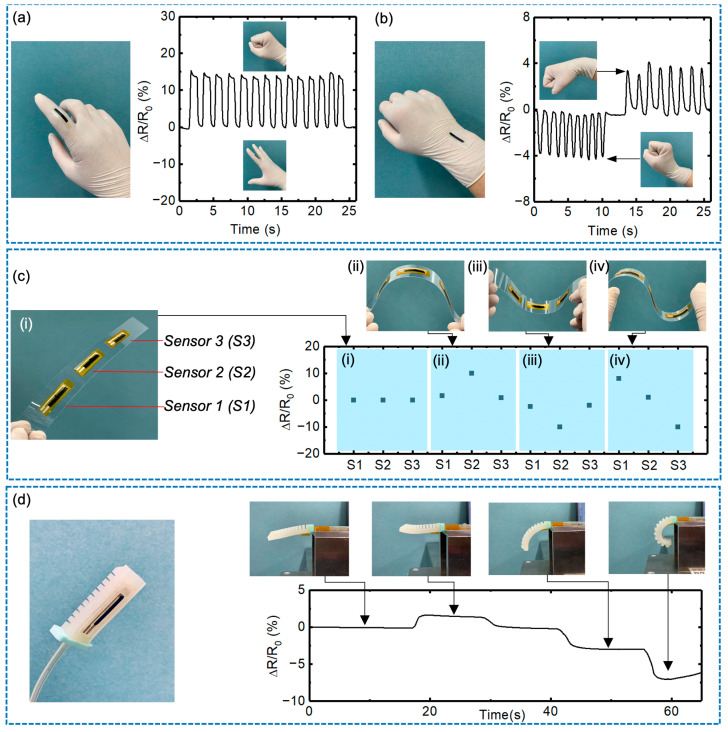
(**a**) Photograph of the printed bending sensor attached to a human’s index finger and the sensor’s resistance changes with different hand gestures and bending states. (**b**) Photograph of the printed sensor attached to a human wrist and the sensor’s resistance changes with wrist motions. (**c**) Photograph of multiple printed sensors attached to a flexible substrate and the resistance of sensors with different shapes. (**d**) photograph of a soft robot gripper integrated with a bending sensor and the sensor’s resistance changes in different actuating states.

**Table 1 sensors-23-05041-t001:** Sensor performance comparisons between this work and the recently reported studies.

Materials	Tensile *GF*	Compressive *GF*	Method	Reference
Carbonized cellulose	10.1	4.45	creping	[34]
Leather/CNT	5.68	12.56	drop casting	[36]
MWCNT	13. 07	12.87	printing	[37]
CNT/CB/paper	7.5	2.6	dip-coating	[38]
RGO/paper	7.99	18.96	soaking	[39]
AgNW/CNF	10.2	1.2	filtration	[40]
PDMS/CB	35.2	45.6	printing	this work

CNT: carbon nanotube; RGO: reduced graphene oxide; PEDOT/PSS NW: poly(3,4-ethylenedioxythiophene)-poly(styrenesulfonate) nanowire; AgNW: silver nanowire; CNF: cellulose nanofiber.

## Data Availability

Data sharing is not applicable to this article.

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
