# Peer review of "Printed Directional Bending Sensor with High Sensitivity and Low Hysteresis for Human Motion Detection and Soft Robotic Perception"

_sensors, 2023, doi:10.3390/s23115041_

Round 1

Reviewer 1 Report

In this manuscript, a high-performance flexible bending sensor has been developed using a printable porous conductive composite composed of PDMS and carbon black. This simple and scalable sensor demonstrates bidirectional high sensitivity, negligible hysteresis, good linearity, and excellent durability. The porous structure and segregation between PDMS and CB in the composite enable superior mechanical-electric performance compared to conventional random composites.

The development and application of this material is really really interesting and fascinating. The authors have provided a thorough and detailed analysis of their research, which is presented in a clear and concise manner and the visual representation of the data is both informative and visually appealing.

However, since this material and the fabrication process has been mentioned in the previous article [30, 31], although not as detailed as this manuscript and its supplementary material, I don't quite understand the main purpose of this article.

In the manuscript, the author said: “In this study, we present the first demonstration of a fully stencil-printed flexible directional bending sensor using this self-segregated porous conductive composite, and thoroughly examine its electromechanical response to the small strains induced by bending.”
However, the fabrication process has been published in your publication in 2021. Even the sketch of the process description looks very similar, but the pattern of the sensing part is changed. The electromechanical performance also has been tested and shown in your publication in 2023. Could the authors please provide a clear explanation of the differences and how this work contributes to the existing research?

Reviewer 2 Report

Thank you for submitting this manuscript.

The paper describes a novel bending strain sensor based on a PDMS/CB composite that attains superior performance in comparison to state of the art due to its porous structure with conductive fillers lining the pore walls. The methods and experiments setups are well described and conclusions are supported by relevant data.

While I do recommend this paper for publication, I have some minor suggestions/comments that would be helpful to improve the readers understanding of the paper.

1) In lines 83-85 and 88-90, could you please quantify the amount of carbon black being added to the solution? This will help fellow researchers reproduce the experiments detailed in this paper.

2) Can the authors comment on possible use of this sensor to detect torsional strain? Can these sensors be used to detect a twisting motion of a surface?

3) What is the future scope of work with this strain sensor? What area does the sensor currently lack in that needs further development?

Reviewer 3 Report

The authors in this manuscript developed the banding sensor using CB, PDMS, and DES. It is very impressive to envision a porous conductive composite using the spontaneous phase separation of PDMS and CB. Moreover, it is considered practical because the production process is simple, utilizing the stencil printing method, and using inexpensive materials with the industrial in mind. However, the ambiguous statements in the manuscript raise some questions. In order to improve the manuscript, it is highly recommended that the authors check several suggestions mentioned below.

1.     There seem to be some parts that need English correction, such as some English conjunctions inconsistency, so it would be good to proceed with English correction.

2.     In part 3.2, conductivity has a related figure, however, there is no mention of data on percolation threshold.

3.     Line 177. Why does the sensitivity decrease as a result of reducing the void space? Please explain additional statements in the manuscript.

4.     It can be explained that the carbon black in the random composite exhibits lower sensitivity at a higher content of 5wt%. However, this may be less intuitive because the comparison is being made to the 2wt% and 4wt% levels of the porous composite.

5.     It should be noted which wt% each description in figure 3 is targeting.

6.     In order to ensure a coherent argument, it is recommended to replace the phrase "an ignorable hysteresis" with "negligible hysteresis" in line 237-238. This change will improve the clarity and accuracy of the statement.

7.     Line 259-261. It is necessary to comment or evaluate whether the proposed response and recovery time are superior compared to other sensors. Line 295-296. The benefits that can be claimed with the data in figure 4c deserve further explanation.

8.     It would be beneficial to include data for the 2wt% and 4wt% Random composites in Figure S4. This addition can provide a more complete comparison between the Random and Porous composites and allow for a better understanding of the behavior of the sensor at different composite compositions.

 By addressing these suggestions, the authors can improve the clarity and accuracy of their manuscript and enhance its potential impact in the field of sensor technology.

Round 2

Reviewer 1 Report

The author's response provides a clear explanation of the importance and value of this study. However, the differences between this study and the previous two published works were still not explicitly explained in the revised manuscript, except for the improvement in the detection range. This study used similar experimental methods and procedures used in the previous studies, the novelty of this manuscript seems a little bit lacking. It would be helpful if the author could provide more specific details about the other contributions of this study beyond the detection range improvement.
